# Scaled Model for Studying the Propagation of Radio Waves Diffracted from Tunnels

Ori Glikstein [1], Gad A. Pinhasi [2],* and Yosef Pinhasi [1]

1   Department of Electrical and Electronic Engineering, Ariel University, Ariel 40700, Israel;
    orig@ariel.ac.il (O.G.); yosip@ariel.ac.il (Y.P.)
2   Department of Chemical Engineering, Ariel University, Ariel 40700, Israel
*   Correspondence: gadip@ariel.ac.il

**Abstract:** One of the major challenges in designing a wireless indoor–outdoor communication network operating in tunnels and long corridors is to identify the optimal location of the outside station for attaining a proper coverage. It is required to formulate a combined model, describing the propagation along the tunnel and the resulting diffracted outdoor pattern from its exit. An integrated model enables estimations of the radiation patterns at the rectangular tunnel exit, as well as in the free space outside of the tunnel. The tunnel propagation model is based on a ray-tracing image model, while the free-space diffraction model is based on applying the far-field Fraunhofer diffraction equation. The model predictions of sensing the radiation intensity at the tunnel end and at a plane located at a distance ahead were compared with experimental data obtained using a down-scaled tunnel model and shorter radiation wavelength correspondingly. This down-scaling enabled detailed measurements of the radiation patterns at the tunnel exit and at the far field. The experimental measurements for the scaled tunnel case fit the theoretical model predictions. The presented model accurately described the multi-path effects emerging from inside the tunnel and the resulting outdoor diffracted pattern at a distance from the tunnel exit.

**Keywords:** diffraction; indoor–outdoor propagation; ray tracing; tunnel path loss; wireless communication

## 1. Introduction

Indoor and outdoor radio signal propagation modeling plays an important role in designing wireless communication systems. In recent years, there has been increasing interest in the characterization of radio-wave propagation in long corridors and tunnels [1,2]. Theoretical and experimental studies have been directed toward finding the link budget along tunnels to evaluate the expected wireless link performance for different frequencies [3]. Experiments have been performed in car and train tunnels [4,5], as well as in rectangular and curved tunnels [6].

Samad et al. [7] presented a survey of various techniques that are used to estimate path loss for wireless communication inside tunnels. In this study, wave propagation models for tunnels were analyzed in depth and evaluated qualitatively regarding their notable features, characteristics, competitive advantages, and limitations. Although many methods have been developed, ray-tracing and modal methods are the two major analytical approaches for modeling radio propagation in straight tunnels.

Zhou et al. [8–10] modeled radio-wave propagation in straight tunnels using ray tracing. The effect of tunnel wall roughness on wave propagation was studied via combining the ray-tracing model with the waveguide modal approach. It was found that surface roughness in tunnels introduces additional attenuation to RF signals that decreases with tunnel dimensions rapidly and linearly increases with wavelength [9].

Hossain et al. [11] studied indoor radio propagation using the three-dimensional ray-tracing (ETRT) method. Zhao et al. studied the effects of antenna location and polarization on radio-wave propagation in tunnels [12]. They employed an image-based ray-tracing

method to model radio-wave propagation in tunnels and proposed an optimized scheme for the spatial and polarization diversities of tunnel antennas [13].

Oladimeji et al. [14] present a survey of various propagation models and measurement work conducted in millimeter waves for the indoor environments. An analysis of path loss and shadow fading in different frequency bands for 5G networks is presented.

Millimeter-wave propagation models are being used to study the path loss across different network parameters, environments, and terrain (rural, suburban, urban, and urban high-rise) for cellular and satellite networks [15–19].

Theoretical and experimental studies [20–23] relate to power losses, phase dispersion, and temporal delay spread. In these studies, experimental tests were performed using full-scale tunnels and a scaled model with corresponding conditions.

Wave diffraction out from a tunnel can be addressed as diffraction at the open-ended circular waveguide. Galyamin et al. [24,25] solved the problem of diffraction at the open-ended circular waveguide for the case of dielectric-loaded [24] and dielectric-lined [25] waveguides. They proposed an analytical model that describes the interaction of the EM waves (free or guided) by solving the Helmholtz equation in the frequency domain. The model equation was solved numerically.

Although extensive studies have been conducted to analyze wave propagation across tunnels, there is a lack of information on the combined propagation of radiation outside of the tunnel. This information is essential for indoor–outdoor communication design.

The current paper presents a theoretical analysis of radio-wave propagation along tunnels and outside the tunnel exit. The wave propagation in a tunnel is calculated using a ray-tracing propagation model of the propagation outside of the tunnel using far-field diffraction model. This theoretical study is verified experimentally by analyzing radiation in a scaled-down tunnel structure. This down-scaling enables detailed measurements of the radiation patterns at the tunnel exit and at the far field, while experiments in such resolutions in a real tunnel would not be realistic because of the large dimensions and distances. Down-scaling requires shortening the wavelength correspondingly by the same factor.

A comparison is made between the calculated results and the experimental field intensity-sensing results for the tunnel cross-section aperture as well as for the outdoor far-field zone, away from the tunnel exit.

This paper also designates a comfortable and efficient technique for the experimental validation of theoretical estimations, by using a down-scaled compact model, while utilizing shorter wavelengths (higher frequencies) correspondingly.

The rest of the paper is organized as follows: Section 2 describes in detail the indoor–outdoor propagation model in two stages: the propagation within the tunnel and the diffraction in the free space. Section 3 presents the experimental setup for the full-scale and down-scaled tunnel model. Section 4 presents the numerical and experimental results for radiation intensity patterns at the tunnel exit and at a distance from the exit plane. Finally, Section 5 concludes the paper.

## 2. The Indoor–Outdoor Model

The indoor–outdoor propagation model is aimed at calculating the electromagnetic radiation transferred through a straight tunnel in an open-ended rectangular waveguide and diffracted from its outlet aperture to a plane located at a distance $d$ from the exit plane, as can be seen in Figure 1 [23].

Let us consider a tunnel as a straight hollow dielectric waveguide with rectangular cross-sectional dimensions ($2a \times 2b$). The origin of the coordinate system is in the center of the waveguide cross-section, with $x$ being horizontal, $y$ being vertical, and $z$ being along the tunnel. The tunnel starts at Plane 0, with coordinates ($x_0, y_0, z_0$), the tunnel exit is at a length $L$ (Plane 1: $x_1, y_1, z_1$), and the outdoor illumination plane is at a distance d from the tunnel exit (Plane 2: $x_2, y_2, z_2$).

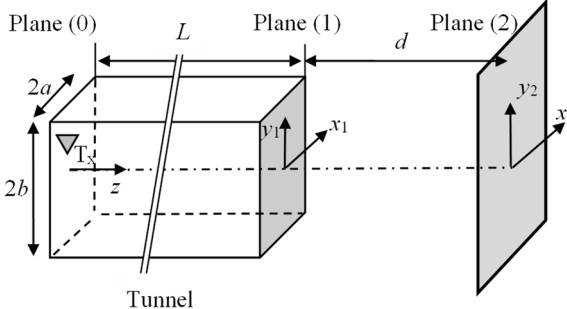

**Figure 1.** Tunnel geometry, transmitter (T$_x$) (Plane 0), tunnel exit (Plane 1), and diffraction plane (Plane 2).

The transmitter is located on plane 0 with arbitrary coordinates of T$_x$ ($x_0$,$y_0$,$z_0$) and the sensor is located on plane 2 with arbitrary coordinates of R$_x$ ($x_2$,$y_2$,$z_2$).

The model calculates the radiation intensity at the tunnel exit (Plane 1: $z_1 - z_0 = L$) and at a distance $d$ from the tunnel exit (Plane 2: $z_2 - z_1 = d$). It contains two sub-models: the first is "the tunnel model", which calculates the radiation field at the tunnel exit (Plane 1) based on the ray-tracing model, and the second is "the outdoor diffraction model", which calculates the radiation field at distance $d$ from the tunnel exit based on the Fraunhofer diffraction model. The model structure scheme is presented in Figure 2.

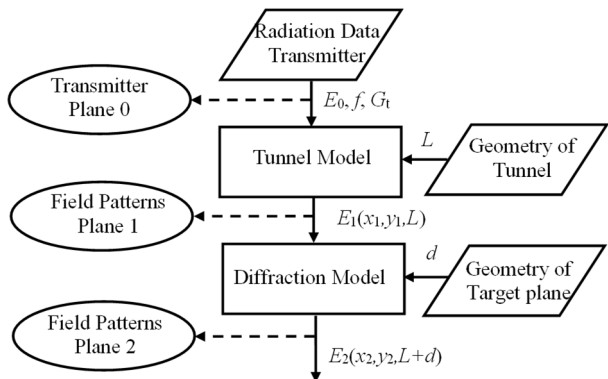

**Figure 2.** Flow diagram of the modeling scheme.

### 2.1. Indoor Multi-Ray Propagation Model

The "ray-tracing method" is based on presenting the electromagnetic (EM) field as a large ensemble of very narrow beams (rays) propagating in straight lines. Thus, the reflection, diffraction, and scattering effects on the wavefront are approximated using simple geometric equations instead of by solving Maxwell's wave equations.

The method uses the Free-Space Path Loss model, using the generalized "Friis free-space equation" combined with multiple reflections. The reflected field can be calculated by multiplying the incident field by the corresponding Fresnel reflection coefficient [1,8,9].

Based on the ray-tracing method, the electric field reflected rays in a hollow tunnel can be represented by a summation of the electric fields at images of the transmitter (T$_x$) in the tunnel inlet plane [9]. The line-of-sight ray, reflected ray, and image index ($n$,$m$) maps can be seen in Figures 3 and 4. The model presented in the current section is based on the image ray-tracing model [1,8] with additional modification to include the case of walls with different properties.

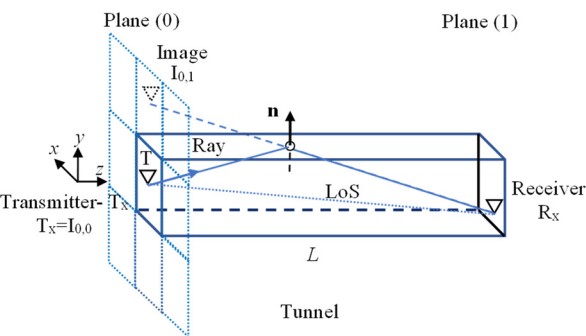

**Figure 3.** Tunnel geometry and refraction model: ray source-image line ($r_{mn}$).

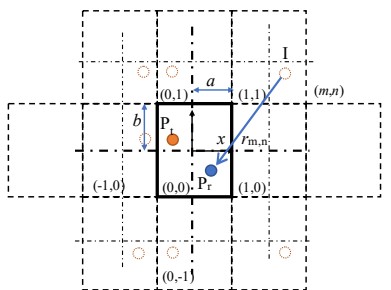

**Figure 4.** Tunnel transmitter plane and refraction model: ray source-image projection ($r_{mn}$). Transmitter point ($P_t$-●), Receiver point ($P_r$-●), Transmitter image (I ○).

The electric field at an arbitrary point $R_x$ ($x_1,y_1,z_1$), within the tunnel, $\tilde{\mathbf{E}}_{r,1}(x_1,y_1,z_1)$, can be expressed as a function of the transmitted electric field, $\tilde{\mathbf{E}}_t(x_0,y_0,z_0)$, from the transmitter point $T_x$ ($x_0,y_0,z_0$) at Plane 1:

$$\frac{\tilde{\mathbf{E}}_{r,1}(x_1,y_1,z_1)}{\tilde{\mathbf{E}}_t(x_0,y_0,z_0)} = \left(\frac{\lambda}{4\pi}\right) \sum_{m=-\infty}^{\infty} \sum_{n=-\infty}^{\infty} \frac{G_{m,n}}{r_{m,n}} \rho_{m,n} e^{-j\frac{2\pi f}{c}r_{m,n}}, \tag{1}$$

where $\lambda = c/f$ is the wavelength, $f$ is the frequency, and $c = 2.998 \times 10^8$ m/s is the speed of light. The integers $m$ and $n$ represent the number of reflections that the ray undergoes relative to the horizontal and vertical walls, respectively. The antenna gains for the transmitter—$t$ $G_t$ and receiver—$r$ $G_r$ are functions of the spatial direction and antenna polarity. The total gain corresponding to path $m,n$ is $G_{m,n} = \sqrt{G_r G_t}$.

For a tunnel with four walls with different properties (Figure 5), the total reflection coefficient for a reflected ray can be written as

$$\rho_{m,n} = \rho_U^{|n_U|}\rho_D^{|n_D|}\rho_R^{|m_R|}\rho_L^{|m_L|}, \tag{2}$$

where $\rho_i$ is the reflection coefficient for each wall $i$ ($i=$ up—$U$, down—$D$, right—$R$, left—$L$), and $m_i$ and $n_i$ are the number of reflections for each wall:

$$\begin{matrix} m_R = \lceil m/2 \rceil & m_L = \lfloor m/2 \rfloor \\ n_U = \lceil n/2 \rceil & n_D = \lfloor n/2 \rfloor \end{matrix}, \tag{3}$$

where $\lceil x \rceil$ represents rounding toward positive infinity (ceiling) and $\lfloor x \rfloor$ represents rounding toward negative infinity (floor). The total number of reflections from the vertical wall is ($n_U + n_D = n$) and that from the horizontal wall is ($m_R + m_L = m$).

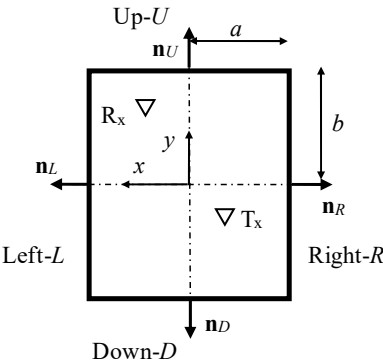

**Figure 5.** The tunnel's cross-sectional geometry and the orientation of its walls and their normal vectors.

The ray vector between the transmitter image $I_{m,n}$ $(x_m, y_n, z_t)$ and the receiver point $R_x$ $(x_1, y_1, z_1)$ is represented as

$$\mathbf{r}_{m,n} = (x_1 - x_m)\hat{\mathbf{x}} + (y_1 - y_n)\hat{\mathbf{y}} + (z_1 - z_t)\hat{\mathbf{z}}, \tag{4}$$

where $(m,n)$ are image indexes in the $x$ and $y$ directions, respectively. The length of the image ray is

$$r_{m,n} = |\mathbf{r}_{m,n}| = \sqrt{(x_1 - x_m)^2 + (y_1 - y_n)^2 + (z_1 - z_t)^2}, \tag{5}$$

where the receiver image position, $I_{m,n}$ $(x_m, y_n, L)$, is

$$x_m = 2mb + (-1)^m x_0, \quad y_n = 2na + (-1)^n y_0. \tag{6}$$

The line of sight is obtained for $m = n = 0$: $r_{0,0} = r_{LOS}$.

By defining the surface direction by a unit normal vector pointing outside of the tunnel, $\mathbf{n}$, the incidence angle is obtained as follows:

$$\cos(\theta_{m,n}) = \frac{\mathbf{r}_{m,n} \circ \mathbf{n}}{r_{m,n}}, \tag{7}$$

where $\theta$ is the angle of incidence and $\varphi_g = \pi/2 - \theta$ is the grazing angle.

For the rectangular channel, the incidence angle relative to a unit normal vector to the reflecting surface is

$$(\theta_{m,n})_{R,L} = \cos^{-1}\left(\frac{|x_r - x_m|}{r_{m,n}}\right), \quad (\theta_{m,n})_{U,D} = \cos^{-1}\left(\frac{|y_r - y_n|}{r_{m,n}}\right), \tag{8}$$

where $\theta_{U,D}$ and $\theta_{L,R}$ are the vertical (side walls: $R$, $L$) and horizontal (top and bottom: $U$, D) incidence angles, respectively.

For a plane-wave incident on a smooth surface, it is known that the wave is reflected in the specular direction, given by Snell's law of reflection. The reflected field can be calculated by multiplying the incident field with the corresponding Fresnel reflection coefficient $\rho$, given by calculating

$$\rho_{\perp,\parallel} = \frac{\cos\left(\theta_{\perp,\parallel}\right) - \Delta_{\perp,\parallel}}{\cos\left(\theta_{\perp,\parallel}\right) - \Delta_{\perp,\parallel}}, \tag{9}$$

where

$$\Delta_\perp = \sqrt{\bar{\varepsilon}_i - \sin^2(\theta)}, \quad \Delta_\parallel = \sqrt{\bar{\varepsilon}_i - \sin^2(\theta)}/\bar{\varepsilon}_i, \tag{10}$$

where $\rho$ is the reflection coefficient, $\theta$ is the incidence angle relative to a unit normal vector to the reflecting surface, and $\Delta$ is a quantity related to the effective surface impedance. Here, the subscripts $\perp$ and $\parallel$ denote the reflection modes where the E-field component is normal to the plane of incidence, i.e., the TE mode (s-mode), or where the E-field component is

in the plane of incidence, i.e., the TM mode (p-mode), respectively. For a long rectangular channel, the reflection modes are approximated from the antenna polarity (V or H) and the wall orientation (*U,D,R,L*) (see Table 1 and Figure 5).

**Table 1.** Polarity at the tunnel walls.

| Wall | Normal Vector | Incidence Angle | Wave Polarity | |
|---|---|---|---|---|
| | | | Vertical, V | Horizontal, H |
| Top/bottom: (*i* = *U, D*) | $\mathbf{n} = \pm\hat{\mathbf{y}}$ | $\theta_i$ | TM (∥) | TE (⊥) |
| Side walls: (*i* = *R, L*) | $\mathbf{n} = \pm\hat{\mathbf{x}}$ | $\theta_i$ | TE (⊥) | TM (∥) |

The model can account for a rectangular tunnel made of walls with different material properties. The complex relative permittivity of a wall is $\bar{\varepsilon}_i$ ($\bar{\varepsilon}_i = \varepsilon_i/\varepsilon_0$), where (*i* = *U,D,R,L*) denotes the wall orientation, and $\varepsilon_0$ and $\varepsilon_i$ are the permittivity of the air and wall, respectively. The magnetic permeability of the free media is assumed to be the same and equal to that of the free space.

The model can account for different source polarizations—vertical, V or horizontal, H—and antenna types: omnidirectional antennas, horn antennas (conical or pyramidal) [21], and antennas with a given gain function: $G_{r/t}(\theta,\phi)$ [20].

The received signal field (1) can be written relative to the LOS signal field as

$$\frac{\tilde{\mathbf{E}}_r(x_1,y_1,z_1)}{\tilde{\mathbf{E}}_{LOS}(x_1,y_1,z_1)} = \sum_{m=-\infty}^{\infty}\sum_{n=-\infty}^{\infty}\frac{r_{0,0}}{r_{m,n}}\frac{G_{tot}}{G_{LOS}}\rho_{m,n}e^{-j2\pi f\Delta\tau_{m,n}}, \tag{11}$$

where the delay spread of any reflected ray, $\Delta\tau_{m,n}$, is equal to the delay between the LOS ray and the reflected ray:

$$\Delta\tau_{m,n} = \frac{r_{m,n} - r_{0,0}}{c} \tag{12}$$

The received–transmitted power ratio at the point $(x_1,y_1,z_1)$ at the tunnel exit plane is

$$\frac{P_r}{P_t} = \left|\frac{\tilde{\mathbf{E}}_{r,1}(x_1,y_1,z_1)}{\tilde{\mathbf{E}}_t(x_0,y_0,z_0)}\right|^2 \tag{13}$$

where $P_t$ and $P_r$ are the transmitted and the received power, respectively.

The ray-tracing method assumes a finite number of reflectors with known locations and dielectric properties. An error analysis and an analysis of the sensitivity to the number of reflectors calculated in the model were performed in Rapaport et al.'s study [20]. The root-mean-squared error (RMSE) for the *n*-th additional ray was calculated to find that the relative error decreased rapidly with an increase in the number of reflections and tunnel length.

## 2.2. Outdoor Diffraction from Tunnels

The diffraction model calculates the field illuminated on a parallel plane at a distance *d* from the tunnel exit. The Fraunhofer diffraction equation is used to model the diffraction of waves when the diffraction pattern is viewed at a long distance from the diffracting object (tunnel exit).

The field at a distance *d* from the tunnel exit (Figure 6) is given in the far field in Cartesian coordinates as (Plane 2: *x,y*), under the assumption of a relatively large plane with no effects or reflection from the boundaries, as follows [26]:

$$\tilde{\mathbf{E}}_2(x_2,y_2,d) = -\frac{1}{j\lambda d}e^{-j\frac{\pi}{\lambda d}(x_2^2+y_2^2)}\int_{-b}^{+b}\int_{-a}^{+a}\tilde{\mathbf{E}}_1(x_1,y_1)e^{-j\frac{2\pi}{\lambda d}(x_2x_1+y_2y_1)}dx_1dy_1, \tag{14}$$

here, $\tilde{\mathbf{E}}_1(x_1, y_1)$ is the field pattern at the tunnel exit (Plane 1), and $\lambda = c/f$ is the radiation wavelength.

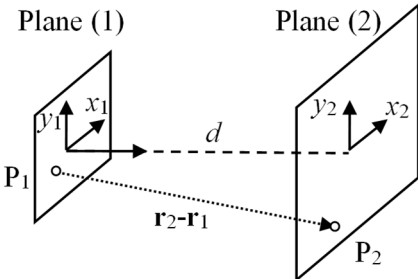

**Figure 6.** Outdoor diffraction pattern at a distance from the tunnel.

The far-field condition that arises from the Rayleigh criterion is expressed via the Fresnel number *F* as

$$F = \frac{s^2}{\lambda d} \ll 1, \tag{15}$$

where the characteristic length of the tunnel plane is $s^2 = (2a)^2 + (2b)^2$. In this study, we assume that Plane 2 is infinite and there are no multi-path effects emerging from additional reflections (such as from the ground).

### 2.3. Numerical Procedure

Both models were solved numerically to find the field distribution at the tunnel ends and at a parallel plane at a distance from the exit.

The propagation model calculated the field at Plane 1. The tunnel exit plane coordinates $(x, y)$ were discretized to $(M, N)$, an odd number of points. The complex field at point $(x_j, y_i)$ was calculated using Equation (1) to obtain the field matrix value $(\mathbf{E}_1)_{ij}$ for a two-dimensional grid of $M \times N$ points.

The diffraction model takes the field distribution at the tunnel exit (Plane 1) from the "tunnel model" as an input and finds the radiation patterns at a distance *d* (Plane 2) via numerical integration of the Fraunhofer diffraction Equation (14).

The double integral was solved numerically using a two-dimensional version of Simpson's 1/3 rule [27]. The numerical integral was written as

$$\begin{aligned}
\tilde{\mathbf{E}}_2(x_2, y_2) &= \int\limits_{-b}^{+b} \int\limits_{-a}^{+a} \tilde{\mathbf{E}}_1(x_1, y_1) e^{-j\frac{2\pi}{\lambda d}(x_2 x_1 + y_2 y_1)} dx_1 dy_1 \approx \\
&\simeq \frac{\Delta x \Delta y}{9} \sum_{i=1}^{N} \sum_{j=1}^{M} W_{ij} E_{1,ij} e^{-j\frac{2\pi}{\lambda d}(x_2(x_1)_j + y_2(x_1)_i)}
\end{aligned}, \tag{16}$$

where the size of the 2D integration element is $\Delta x = 2a/(M-1)$; $\Delta y = 2b/(N-1)$, and $W_{ij}$ is the two-dimensional Simpson's coefficient matrix (the scalar product of two Simpson's coefficient vectors). The Simpson's coefficient matrix is

$$W_{N,M} = V_N^T \circ V_M, \tag{17}$$

where $V_k$ is in a one-dimensional Simpson's coefficient row vector with *k* elements in the form of [1, 4, 2, 4,..., 2, 4, 1].

### 2.4. Uniform Illumination: Test Case

The numerical calculation was tested against an analytical solution of Fraunhofer diffraction patterns from uniformly illuminated square output apertures. For the theo-

retical case of a uniform field ($\tilde{\mathbf{E}}_{10}$) at a rectangular $2a \times 2b$ tunnel exit (Plane 1) [24], the calculation is

$$\tilde{\mathbf{E}}_1(x_1, y_1) = \tilde{\mathbf{E}}_{10}\mathrm{rect}\left(\frac{x_1}{2a}\right)\mathrm{rect}\left(\frac{y_1}{2b}\right), \tag{18}$$

where the rectangular function is

$$\mathrm{rect}(t/T) = \begin{cases} 1 & |t| < T/2 \\ 0 & |t| > T/2 \end{cases}. \tag{19}$$

The resulting field at a distance $d$ (at Plane 2) is

$$\tilde{\mathbf{E}}_2(x, y, d) = -\tilde{\mathbf{E}}_{10}\frac{4ab}{j\lambda d}\mathrm{sinc}\left(\frac{2a}{\lambda}\frac{x}{d}\right)\mathrm{sinc}\left(\frac{2b}{\lambda}\frac{y}{d}\right)\exp\left(-j\frac{\pi}{\lambda d}\left(x^2 + y^2\right)\right), \tag{20}$$

where $\mathrm{sinc}(t) = \sin(\pi t)/(\pi t)$.

Holding the paraxial approximation in the far field, the field can be expressed in terms of the lateral angle $\theta_x = x/d$ and the elevation angle $\theta_y = y/d$. In the uniform illumination case, the expected angular main-lobe width due to the field diffraction from the tunnel exit rectangular aperture can be evaluated as $\Delta\theta_x = \lambda/a$ and $\Delta\theta_y = \lambda/b$ in the horizontal and vertical directions, respectively.

The radiation power-intensity patterns at the distance $d$ from the uniform illumination square source are presented in Figure 7, with the power distribution map at Plane 2 (Figure 7a) and the power at the horizontal and vertical center axes (Figure 7b). For validation of the numerical model, the results show good agreement between the analytical and numerical solutions.

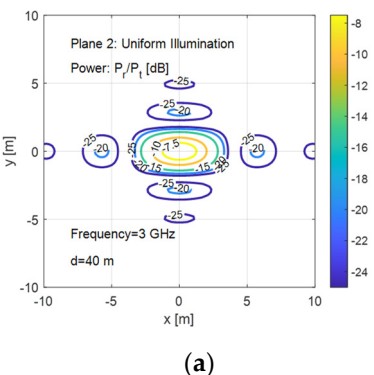
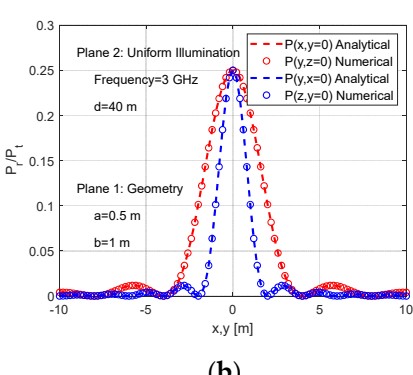

(a)  (b)

**Figure 7.** Outdoor diffraction pattern at a distance from the tunnel for uniform illumination at distance $d = 40$ m: 3 GHz. Vertical polarization (**a**); radiation intensity map (**b**) with radiation intensity on the $x$ and $y$ axes, showing the analytical and numerical results.

## 3. Experimental Setup

The proposed combined model of propagation in a tunnel and diffraction from the tunnel exit was validated experimentally using a full-scale tunnel ("prototype") and its scaled model. The radiation intensity was measured on a plane at the tunnel exit and on a plane at a distance from the exit.

The full-scale tunnel was an underground pedestrian tunnel with a length of $L = 24$ m, a width of $(2a) = 1$ m, and a height of $(2b) = 2$ m, with a radio wave at a frequency of $f = 3$ GHz (or 1 GHz). The scaled-down model of the tunnel had a factor of 10 ($L_{\mathrm{model}}/L_{\mathrm{pro}}$: 1:10). Employing a down-scaled tunnel construction presents an efficient technique for experimental validation of the theoretical estimations, while utilizing shorter wavelengths (higher frequencies) correspondingly. This down-scaling enabled detailed measurements of the radiation patterns at the tunnel exit and in the far field.

The scaled-down construction (Figure 8a) was built out of wood coated with Wood Finish Laminate (Formika), keeping the wall smooth with the value of the complex relative

permittivity ($\overline{\varepsilon}_i$) of the tunnel walls being close to that of the full-scale concrete walls ($\overline{\varepsilon}_i = 5 - 0.85i$) [12]. The thickness of the model wall was 2 cm to prevent sidewall losses.

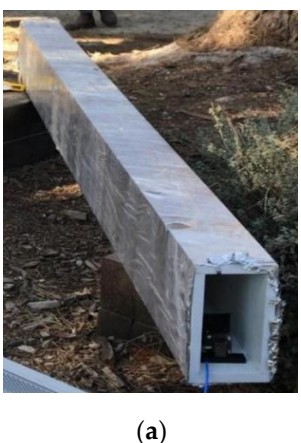

(**a**)

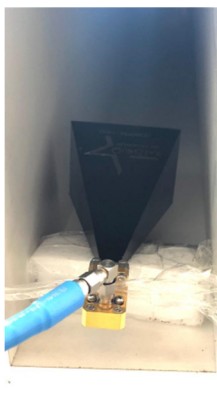

(**b**)

**Figure 8.** Experimental setup; (**a**) scaled tunnel; (**b**) transmitting antenna at the tunnel entrance.

The corresponding scaled model conditions were set using dimensional similitude and analysis. Dimensional analysis was used to express the system with dimensionless parameters: the geometrical similarity, with the scaling ratio $\Pi_{1,i} = x_i/L$, and the wavelength similarity: $\Pi_2 = \lambda/L$. The values of the dimensionless parameters were held to be the same for both the scaled model and the application. By maintaining similarity laws, one can obtain the model testing conditions as

$$\frac{\lambda_{\text{model}}}{\lambda_{\text{pro}}} = \frac{f_{\text{pro}}}{f_{\text{model}}} = \frac{L_{\text{model}}}{L_{\text{pro}}}. \tag{21}$$

The experiments were conducted for two frequencies: 10 and 30 GHz, for horizontal and vertical polarizations. The prototype and the scaled parameters are presented in Table 2.

**Table 2.** Physical parameters and their scaled dimensions.

| Parameter | Normal Vector | Units | Full-Scale Model | Scaled Model |
|---|---|---|---|---|
| Tunnel width | $2a$ | [m] | 1 | 0.10 |
| Tunnel height | $2b$ | [m] | 2 | 0.20 |
| Tunnel length | $L$ | [m] | 25 | 2.5 |
| Frequency | $f$ | [GHz] | 1 and 3 | 10 and 30 |
| Distance from tunnel exit | $d$ | [m] | 40 | 4 |

The experimental setup for the scaled model tests contained transmitter—receiver antennas, a signal generator, and a data analysis system (Rapaport et al., 2020 [20]). The millimeter-wave (MMW) signal was generated using the Keysight N5173B EXG X-Series Microwave Analog Signal Generator (Keysight, Colorado Springs, CO, USA) after multiplication with the Quinstar Active Frequency Multiplier QMM-33142002S (QuinStar Technology, Inc., Torrance, CA, USA). The 30 GHz signal was injected into the tunnel using the Premium Standard Gain Horn QWH-APRS00 (also made by QuinStar) antenna, as shown in Figure 8b. The integrated frequency multiplier and horn antenna are shown in Figure 9.

The pattern of the field intensity at the exit was scanned using a "point" antenna. The localized power was measured using the Rohde-Schwarz FSV40 spectrum analyzer. Scanning was performed with the help of a grid, as illustrated in Figure 10.

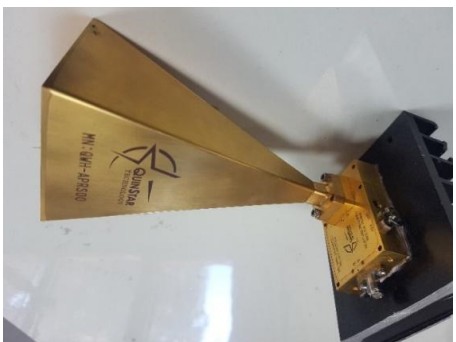

**Figure 9.** The integrated millimeter-wave (MMW) frequency multiplier and directive horn antenna.

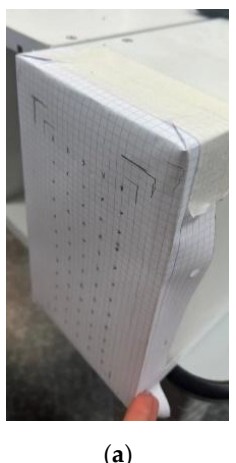

(**a**)

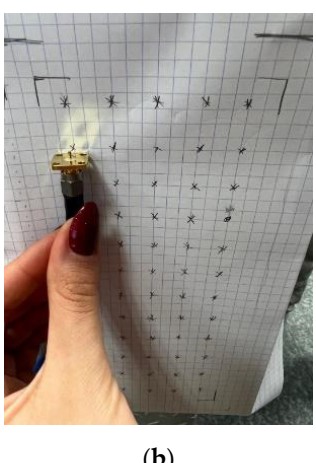

(**b**)

**Figure 10.** Measuring the field pattern at the tunnel exit: (**a**) the grid; (**b**) the receiving-point antenna.

## 4. Results

The propagation in the tunnel and the diffraction to the free space were studied for the full-scale and scaled models. For the two cases, the results of the MMW intensity pattern at the exit of the tunnel and in the free space at a distance from the exit plane are presented. A comparison between the measured and calculated power intensities is presented for the scaled model case.

### 4.1. Modeling Results: Full-Scale Model

For the full-scale case, the model predictions for the radiation intensity are presented at the tunnel exit (Figure 11) and on a parallel plane at a distance $z_2$ = 40 m (Figure 12) for a frequency of 3 GHz and vertical polarity.

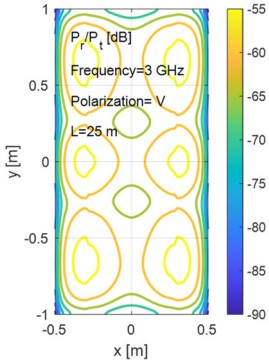

**Figure 11.** Radiation intensity map at the tunnel exit ($L$ = 25 m) in [dB]: full-scale tunnel, 3 GHz, vertical polarization.

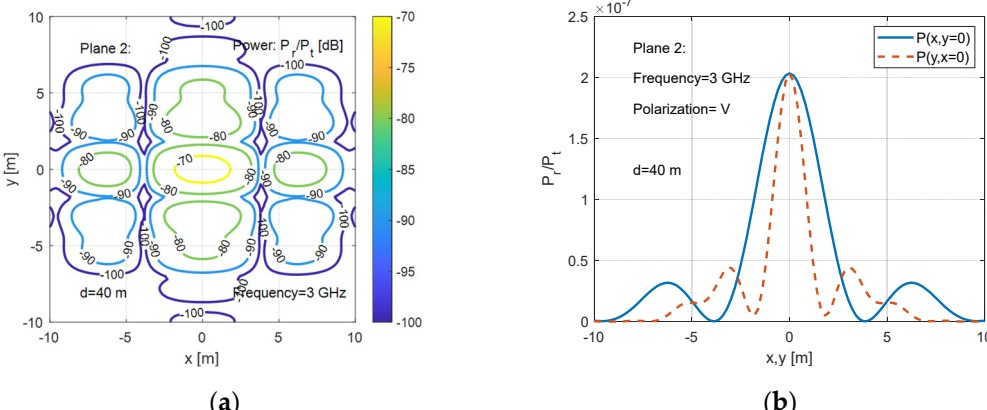

**Figure 12.** Diffraction from the tunnel exit plane at distance $d$ = 40 m. (**a**) Radiation intensity pattern in [dB]; (**b**) radiation intensity at the center axis: $y = 0$ and $z = 0$. Full-scale tunnel.

The results are presented as plane power distribution maps and the power at the center axis for $y = 0$.

The power distribution at the tunnel exit seems to be periodic in nature and is affected by the aspect ratio of the tunnel cross-section ($b/a$). For the diffraction results, the radiation distribution patterns are characterized by images around the line of sight.

### 4.2. Experimental and Modeling Results: Scaled Model—Tunnel Exit

The tunnel propagation model was verified against the experimental results for different frequencies (10 and 30 GHz) and different source polarizations (vertical or horizontal). The results are presented as radiation intensity pattern maps at the tunnel exit plane for the model predictions (a) and experimental measurements (b) (the measurement points are marked in the grid as (X)). See Figures 13–16.

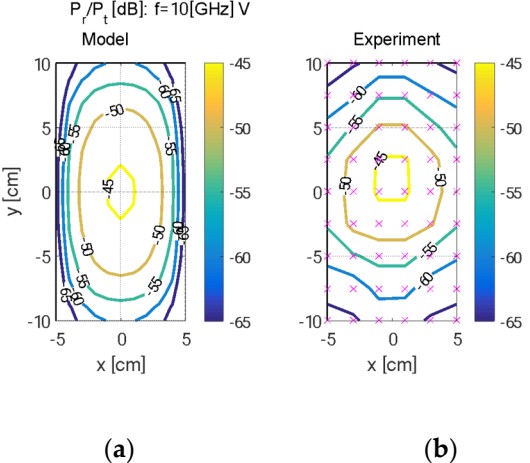

**Figure 13.** Radiation intensity pattern at the tunnel exit plane: 10 GHz, vertical polarization, scaled model tunnel. (**a**) Model prediction; (**b**) experimental results (x—measurement points).

One can see the likeness in the pattern structure between the experimental and modeling results over the entire tunnel exit plane. There are slight differences in the values due to the limited measurement points over the tunnel aperture. Furthermore, there is a good resemblance between the full-scale radiation pattern (Figure 11) and the result of the corresponding scaled model (Figure 15).

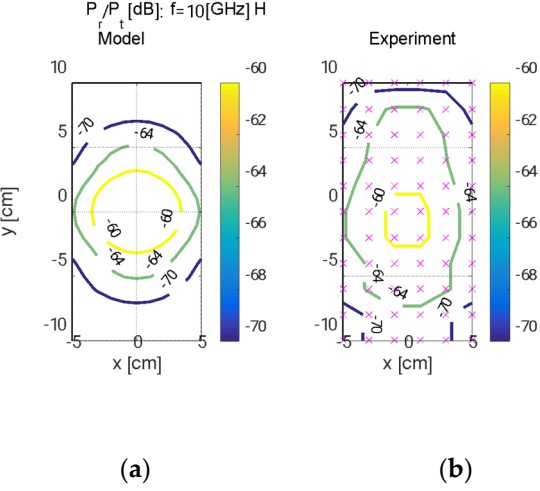

**Figure 14.** Radiation intensity pattern at the tunnel exit plane: 10 GHz, horizontal polarization, model tunnel. (**a**) Model prediction; (**b**) experimental results (x—measurement points).

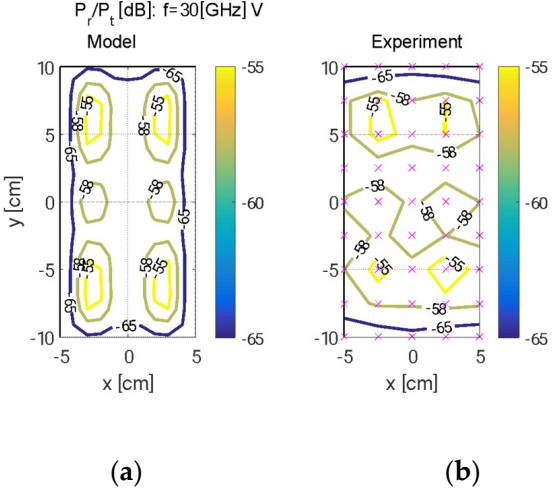

**Figure 15.** Radiation intensity pattern at the tunnel exit plane: 30 GHz, vertical polarization, model tunnel. (**a**) Model prediction; (**b**) experimental results (x—measurement points).

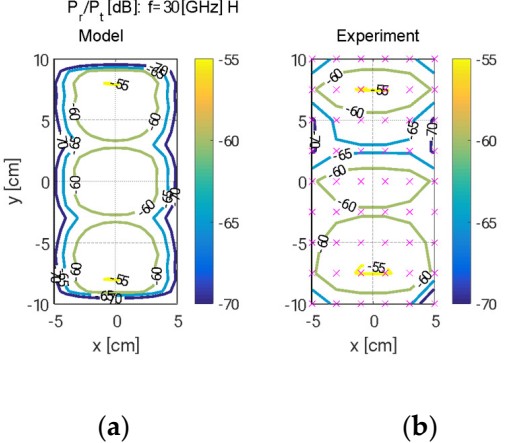

**Figure 16.** Radiation intensity pattern at the tunnel exit plane: 30 GHz, horizontal polarization, model tunnel. (**a**) Model prediction; (**b**) experimental results (x—measurement points).

### 4.3. Experimental and Modeling Results: Scaled Model—Diffraction

The combined propagation and diffraction model was verified against experimental results for the same frequency (30 GHz) and different source polarizations (vertical or horizontal); these results are presented in Figures 17 and 18.

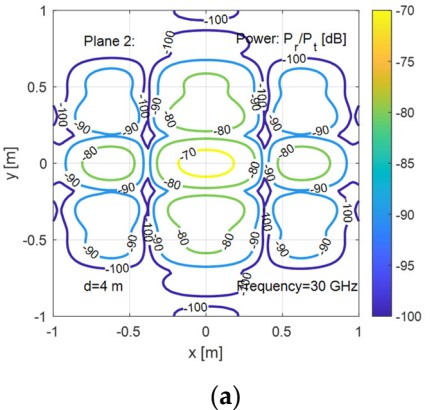

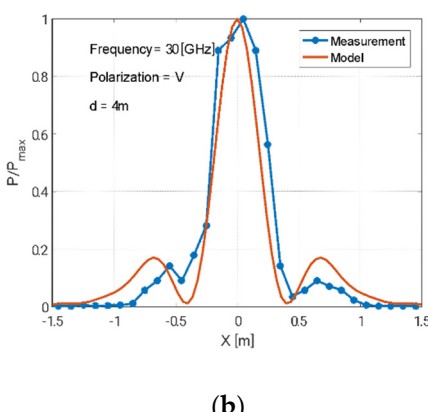

(**a**)

(**b**)

**Figure 17.** Diffraction from the tunnel exit plane at distance $d$ = 4 m: 30 GHz, vertical polarization, model tunnel. (**a**) Radiation intensity pattern; (**b**) measurement results and model prediction with radiation intensity at $y_2$ = 0.

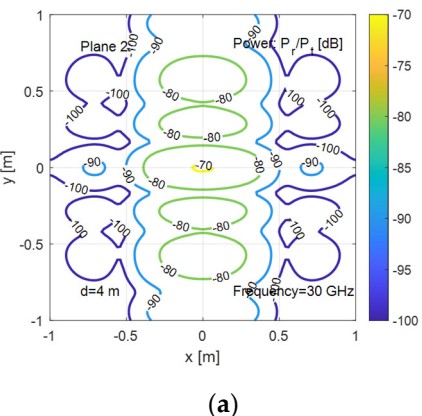

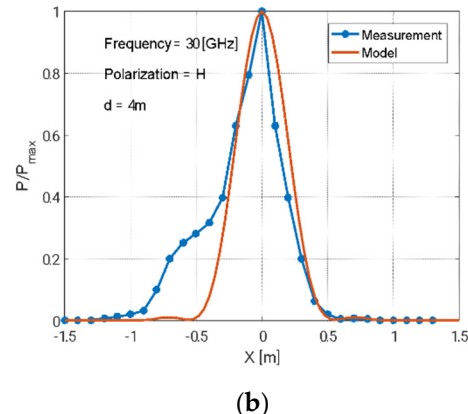

(**a**)

(**b**)

**Figure 18.** Diffraction from the tunnel exit plane at distance $d$ = 4 m: 30 GHz, horizontal polarization, model tunnel. (**a**) Radiation intensity pattern; (**b**) measurement results and model prediction with radiation intensity at $y_2$ = 0. Outdoor diffraction pattern at a distance from a tunnel for uniform illumination.

The model predictions and the experimental results are presented for the scaled model at a distance $d$ = 4 m from the tunnel exit plane. The model predictions are presented as radiation intensity pattern maps in Figure 18a and a comparison against the experimental measurements at the center axis of $y$ = 0 is presented in Figure 18b).

One can see good agreement between the experimental and modeling results at the center axis for both polarization cases. Furthermore, there is considerable resemblance between the full-scale radiation pattern (Figure 12) and the result of the corresponding scaled model (Figure 17).

### 5. Conclusions

In the current study, a combined propagation and diffraction model is presented for estimating the radiation patterns at a rectangular tunnel's exit and in the free space outside of the tunnel. The indoor wave propagation analysis inside the tunnel is based on a quasi-optical ray-tracing method using images, while the outdoor free-space diffraction calculation is based on applying the Fraunhofer equation.

Propagation in the tunnel and the following outdoor diffraction were studied for full-scale and down-scaled tunnels. The experimental measurements of the electromagnetic field patterns at the tunnel exit and at an outdoor distance in free space fit the theoretical predictions resulted from the numerical model. Utilizing a down-scaled model is found to be an effective tool for characterizing wave propagation experimentally, when measurements in real scenarios are not conceivable.

The developed models enable calculations of the following radiation field properties: intensity, phase, and time delay. This study reveals the important mechanisms that dominate wave diffraction from the exits of tunnels and long corridors. They can therefore serve as reliable computational tools for evaluating the expected performances of remote-sensing schemes and wireless communication links. The model was found to accurately describe the multi-path effects emerging from inside the tunnel and the resulting outdoor diffracted pattern at a distance from the tunnel exit.

**Author Contributions:** Project design and manuscript preparation, O.G., G.A.P. and Y.P.; code writing and analysis, G.A.P.; methodology, experimental study, and analysis, O.G.; code writing, G.A.P.; supervision, Y.P. All authors participated in finalizing the manuscript. All authors have read and agreed to the published version of the manuscript.

**Funding:** This research received no external funding.

**Data Availability Statement:** The data presented in this study are all available within this article.

**Conflicts of Interest:** The authors declare no conflicts of interest.

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
