# Peer review of "Scaled Model for Studying the Propagation of Radio Waves Diffracted from Tunnels"

_electronics, doi:10.3390/electronics13101983_

Round 1
Reviewer 1 Report
Comments and Suggestions for Authors
I have the read paper carefully, and I assume it needs some work to be done, please refer to the following comments:
· Abstract: It is not written appropriately. The abstract should be comprehensive, i.e. include purpose, methods and numerical findings.
· In the last paragraph of the introduction dedicate a paragraph that gives the layout of the paper.
· There are many self-citations.
· The literature review part is short, authors are requested to see more recent related studies.
· The study algorithm is good, the mathematical model is supported by experimental work. However, the experimental study is not related to real life scenario, in other words, the experiment should have been done in a real tunnel, while the current experiment is a demo of waveguide propagation.
· Comparison should been done with other published related work and highlight the significant of your model.
· What is the purpose of selecting this Operating frequency range?
· What are the material components of the experimental wall? And what is the thickness of the walls.
· There are already existing models that characterize the propagation within tunnels, what makes the proposed model different? If there are reasons, compared results should support the claim.
Comments on the Quality of English Language
Minor revision required.
Author Response
Reviewer 1
Comments and Suggestions for Authors
I have the read paper carefully, and I assume it needs some work to be done, please refer to the following comments:
- Abstract: It is not written appropriately. The abstract should be comprehensive, i.e. include purpose, methods and numerical findings.
Author response:
Following this remark, the abstract is thoroughly revised and extended with the required information.
- In the last paragraph of the introduction dedicate a paragraph that gives the layout of the paper.
Author response:
In accordance with this remark, we add the layout of the paper in the last paragraph of the introduction.
- There are many self-citations.
Author response:
Following this remark, we remove some of the self-citations and add new relevant references.
- The literature review part is short, authors are requested to see more recent related studies.
Author response:
Following this remark, we extend the literature survey and add new related references.
- The study algorithm is good, the mathematical model is supported by experimental work. However, the experimental study is not related to real life scenario, in other words, the experiment should have been done in a real tunnel, while the current experiment is a demo of waveguide propagation.
Author response:
This is important remark !
The study is aimed at presenting an integrated algorithm for describing an indoor-outdoor wave propagation from tunnels to free-space. It also designates a comfortable and efficient technique for experimental validation of the theoretical estimations, by using a downscaled compact model, while utilizing shorter wavelengths (higher frequencies) corresponseingly.
The downscaling enables detailed measurements of the radiation patterns at the tunnel exit and at far field, while experiments in such resolutions in real tunnel is not realistic because of the large dimensions of its aperture and the distances.
The good agreement between the theoretical analysis and the experimental measurements, demonstrates that the integrated algorithm is valid and can be used to forecast the expected link performance in reality.
Following this relevant remark, we add explanations in the introduction and in section 3 describing the experimental setup, clarifying this important issue!
- Comparison should been done with other published related work and highlight the significant of your model.
Author response:
Following this remark, we present a comparison with related work (in the introduction) and emphasize the significance of the model in the abstract and conclusions.
- What is the purpose of selecting this Operating frequency range?
Author response:
Note that we built a downscale construction for verifying the theoretical model experimentally. The downscaling requires shortening the wavelength correspondently by the same factor. In our case, to model a 1mx2m pedestrian tunnel by a 10 times downscale 10cmx20cm model, we need to reduce the wavelength by a factor of 10.
- What are the material components of the experimental wall? And what is the thickness of the walls.
Author response:
The information and consideration were added to the experimental setup description.
- There are already existing models that characterize the propagation within tunnels, what makes the proposed model different? If there are reasons, compared results should support the claim.
Author response:
Certainly, there are models that characterize the propagation inside tunnels. There are mainly to attitudes: ray tracing and waveguide analysis.
Our work is aimed at presenting an integrated model that treats also the resulted propagation outside of the tunnel. Our approach shows that by calculating the radiation pattern at the tunnel exit, one can estimate how the radiation continues to propagate outside the tunnel at free-space. Our theoretical findings are supported by experimental results, employing a convenient setup that enables measurement in high resolution.
Following this relevant remark, we extend the abstract and introduction accordingly, emphasizing our new approach, model and experimental setup.
- Comments on the Quality of English Language
- Minor revision required.
Author response:
The paper was revised to improve the Quality of English Language

Reviewer 2 Report
Comments and Suggestions for Authors
Generally, the authors seem have done a solid work. However, some parts need to be further clarified.
Thus, I suggest major revision with required changes.

Author Response
Reviewer 2
Generally, the authors seem have done a solid work. However, some parts need to be further clarified.
Thus, I suggest major revision with required changes.
This paper developed a model estimating the radiation patterns at the rectangular
tunnel exit as well as at the free space outside of the tunnel. Generally, the authors
seem have done a solid work. However, some parts need to be further clarified. The reviewer has the following concerns:
- In both the title and main text, why the authors described outdoor to indoor
transmission environment as “Open-Ended Tunnels”, which is not commonly used.
Author response:
Following this remark, we changed the terminology accordingly in the revised version.
- In abstract, it is not suggested to use subjective evaluation, such as “Very good
agreement”.
Author response:
Following this remark, we removed all subjective evaluations along the paper.
- The number of the references are out of order, such as from [3] jumps to [6,7].
Author response:
Corrected !
- Both the motivations and contributions need to be further clarified. The authors
should explain why they investigated this paper, and compared this work with the
existing ones in mmWave channel model field to show the novelty of this paper.
Author response:
Following this important remark, the abstract and introduction are thoroughly revised and extended with the required information. Our work is aimed at presenting an integrated model that treats also the resulted propagation outside of the tunnel. Our approach shows that by calculating the radiation pattern at the tunnel exit, one can estimate how the radiation continues to propagate outside the tunnel at free-space. Our theoretical findings are supported by experimental results, employing a convenient setup that enables measurement in high resolution.
Following this relevant remark, we changed the abstract and introduction accordingly, clarifying the motivation and emphasizing our new approach, model and experimental setup
- It is suggested to introduce the following recent work in indoor-outdoor
propagation [R1] and mmWave communications [R2]-[R4] field to highlight the
state-of-art of this paper:
Author response:
The relevant references were included in the introduction and literature review.
- Have the authors considered the penetration loss when the mmWave signal transmit through the blocks between the outdoor and indoor environments?
Author response:
Following this remark, we extend the description of the experimental setup, and clarify that there are no penetration through the tunnel walls.
- Are the conclusions consistent with the evidence and arguments presented? Is there any new finding?
Author response:
Following this remark, the conclusion section was fully revised to include the finding

Reviewer 3 Report
Comments and Suggestions for Authors
In this paper, authors have presented the propagation analysis of diffracted waves from open ended tunnels and corridors at millimeter wave. The computed and measured results are presented and discussed. Following are the suggestions to the authors:
1. In the abstract, use of past sentences such as ‘was developed’ etc is not appropriate.
2. In the introduction section, at some places, the references are cited as ‘Author’s name et al. [X, Y, Z], in this format the first author’s name should be there. For example, Pinhasi is not the first author in all the references [13, 14, 15, 16]. This should be corrected.
3. At the end of introduction section, please include the structure of the paper. Some more recent literature should be included in the introduction.
4. On page 4, Gt and Gr symbols seem different from the equation. Please check.
5. On page 6, in eqn (13) power intensity is denoted by Pr/Pt, please check and correct.
6. As there are two subfigures in Fig 7, the title of these should be given in the title of the figure. The quality of these figures should be improved.
7. In the text, Figure 8 is cited after Figure 9. Please check.
8. In Figs 13-16, please include the subfigure numbering.
9. Please explain the possible reasons for more deviation between the analytical and measured results for the horizontal polarization as compared to the vertical polarization.
10. Conclusion section is a bit long. The small paragraphs should be merged and please avoid citing the references in the conclusion.
11. As suggested above, some more recent references related to the millimeter wave propagation such as ‘Millimeter Wave Channel Characteristics of Outdoor Microcellular based on Improved Ray Tracing Method and BP Neural Network Algorithm’; ‘Propagation Path Loss Prediction Modelling in Enclosed Environments for 5G Networks: A Review’; Improving Path Loss Prediction Using Environmental Feature Extraction from Satellite Images: Hand-Crafted vs. Convolutional Neural Network’ etc should be included.
12. References should be in uniform format. The journal names for some articles are in italic format and for some articles not in italic. Page numbers of ref [12] are missing.
Comments on the Quality of English LanguageMinor editing of English language required.
Author Response
Reviewer 3
(x) Minor editing of English language required
Author response:
The paper was revised to improve the Quality of English Language
Comments and Suggestions for Authors
In this paper, authors have presented the propagation analysis of diffracted waves from open ended tunnels and corridors at millimeter wave. The computed and measured results are presented and discussed. Following are the suggestions to the authors:
- In the abstract, use of past sentences such as ‘was developed’ etc is not appropriate.
Author response:
Following this remark the abstract was changed and corrected accordingally.
- In the introduction section, at some places, the references are cited as ‘Author’s name et al. [X, Y, Z], in this format the first author’s name should be there. For example, Pinhasi is not the first author in all the references [13, 14, 15, 16]. This should be corrected.
Author response:
Corrected !
- At the end of introduction section, please include the structure of the paper. Some more recent literature should be included in the introduction.
Author response:
In accordance with this remark, we add the layout of the paper in the last paragraph of the introduction.
The Literature review was updated accordingly !
- On page 4, Gt and Gr symbols seem different from the equation. Please check.
Author response:
Corrected !
- On page 6, in eqn (13) power intensity is denoted by Pr/Pt, please check and correct.
Author response:
Corrected: The ratio redefined as receiver-transmitter power intensity ratio
- As there are two subfigures in Fig 7, the title of these should be given in the title of the figure. The quality of these figures should be improved.
Author response:
Corrected !
- In the text, Figure 8 is cited after Figure 9. Please check.
Author response:
Corrected !
- In Figs 13-16, please include the subfigure numbering.
Author response:
Done !
- Please explain the possible reasons for more deviation between the analytical and measured results for the horizontal polarization as compared to the vertical polarization.
Author response:
Following this remark, we clarify this point. The study reveals resemblance between the theoretical and experimental pattern structures. There are slight differences in the values due to the limited measurement points over the tunnel aperture.
- Conclusion section is a bit long. The small paragraphs should be merged and please avoid citing the references in the conclusion.
Author response:
Corrected ! The Conclusion section was re-edited.
- As suggested above, some more recent references related to the millimeter wave propagation such as
- ‘Millimeter Wave Channel Characteristics of Outdoor Microcellular based on Improved Ray Tracing Method and BP Neural Network Algorithm’;
- ‘Propagation Path Loss Prediction Modelling in Enclosed Environments for 5G Networks: A Review’;
- Improving Path Loss Prediction Using Environmental Feature Extraction from Satellite Images: Hand-Crafted vs. Convolutional Neural Network’
etc should be included.
Author response:
The relevant references were included in the introduction literature review.
- References should be in uniform format. The journal names for some articles are in italic format and for some articles not in italic. Page numbers of ref [12] are missing.
Author response:
Corrected !
- Comments on the Quality of English Language
- Minor editing of English language required.
Author response:
The paper was revised to improve the Quality of English Language.

Reviewer 4 Report
Comments and Suggestions for Authors
Please see attached file.

Author Response
Reviewer 4
|
It is customary to describe the organization of the paper by sections in the introduction and highlight the contributions of the paper compared to those published in the literature. Author response: In accordance with this remark, we add the layout of the paper in the last paragraph of the introduction. Following this important remark, the abstract and introduction are thoroughly revised and extended with the required information. |
|
To correct: |
|
Maybe is better to write: [8]-[10] and [13]-[16] instead of [8,9,10] and [13,14,15,16]. Author response: Corrected ! |
|
line 67: from the exit plane. (Figure 1) [16]. =>from the exit plane, what can be seen in Figure 1 [16]. (not the period before the parentheses!) Author response: Corrected ! |
|
line 69: Consider => Let consider or We consider Author response: Corrected ! |
|
Change the ordinal numbers in front of the subsection headings: 3.1. Indoor multi-ray propagation model (line 91) => 2.1. 3.1. Outdoor diffraction from tunnels (line 193) => 2.2. 3.1. Numerical Procedure (line 210) => 2.3. 3.1. Uniform Illumination: Test case (line 230) => 2.4. Author response: Corrected !
|
|
Line 92 presenting the electromagnetic (EM) WHAT? Author response: Corrected ! to electromagnetic (EM) field
|
|
Line 98/99 reflection coefficient. [1],[8],[9]. =>reflection coefficient [1], [8], [9]. Author response: Corrected !
|
|
Line 104 [9] (Figure 3, 4). => [9] (Figures 3 and 4). Author response: Corrected
|
|
From the paragraph in lines 102-104 is not clear if the figures 3 and 4 are taken from ref. [9], or they are your own. Take care to be clear if something is taken from the literature. Author response: The figures are original to the current paper and description was added for them.
|
|
Please provide sources for all formulas taken from literature! Now it is not clear which formulas are from the literature and which are yours. Make it clear, and you'll better highlight your contributions. Author response: A clarification note was added for the model source and the original contribution . Sources for the formulas were cited
|
|
Take care of technical aspects of the paper. For example, title of Fig. 3 should be below that figure, not at the next page. Author response: Corrected |
|
Define all variables used in the equations (formulas), as: rm,n and m,n!
Author response: Corrected and all variables in the paper checked to be defined |
|
Figure 5 is too far from the first mention and description. Author response: Corrected
|
|
Figures and should be at the top or bottom of the page as close as possible to the text describing them. Now, Figure 5 is far from the text describing it. Avoid to finish sections with figures! Author response: Corrected
|
|
Line 130: The ray vector between the transmitter image (m,n) Im,n(xm, yn, zt) and the receiver point, Rx(x1, y1, z1). => The ray vector between the transmitter image (m,n) Im,n(xm, yn, zt) and the receiver point, Rx(x1, y1, z1) is: Index t was not defined. Author response: Corrected, the sub symbols transmitter -t and receiver -r were defined
|
|
Line 136: where the receiver image position, Im,n(xm, yn, L): => where the receiver image position, Im,n(xm, yn, L) is: L was not defined. Author response: Corrected: the tunnel exit at a length L |
|
Be sure to define all variables (a, b, c, Pr, Pt, …still are not defined)! Author response: Corrected and defined, with rectangular cross-sectional dimensions (2a×2b) |
|
Put dot or comma at the end of each formula, depend of its position in the sentence! Author response: Corrected |
|
The sentence in line 142 is incomplete. The verb is missing. The situation is the same in many places. Solve this problem! Author response: Corrected and The paper was revised to improve the Quality of English Language
|
|
Use punctuation marks where necessary, e.g. two dots when something follows!!! Author response: Corrected and The paper was revised to improve the Quality of English Language
|
|
Line 160 (U,D,R,L) using Table 1 and Figure 5. => (U, D, R, L) (see Table 1 and Figure 5). Author response: Corrected and
|
|
Edit text from line 161 till 164 on the better way! Author response: Corrected : The petrograph was edited |
|
Line 161 denote the complex relative permittivity => denotes the complex relative permittivity Author response: Corrected
|
|
Line 171 equals the delay => is equal to the delay Author response: Corrected
|
|
Line 271 The frequency used in the model was 30 GHz (or 10 GHz). => The frequencies used in the model was 30 GHz and 10 GHz. Author response: Corrected
|
|
and different source polarizations (vertical or horizontal) (Figures 17, 18). => and different source polarizations (vertical or horizontal), what is presented in Figures 17 and 18. Author response: Corrected
|
|
Line 285 and horn are shown in Figure 8 => and horn antenna are shown in Figure 8 Author response: Corrected
|
|
Two almost identical sentences are given nearby (in lines 271 and 286). Fix it, without repeating the facts at close range.
The frequency used in the model was 30 GHz (or 10 GHz). The experiments were conducted for several frequencies (presented: 10, 30 GHz) for horizontal and vertical polarizations. => The experiments were conducted for two frequencies: 10 and 30 GHz, for horizontal and vertical polarizations.
Author response: Corrected
|
|
Use the writing of the next in the same manner. Now, in line 138 is: line of sight, and in line 312: line-of-sight.
Make the writing of this to be uniform! Author response: Corrected
|
|
See Figure 13-16. => See Figures 13-16.
Author response: Corrected
|
|
According to the Template, references should be written in lowercase letters, except for proper names and abbreviations. Author response: Corrected
|

Round 2
Reviewer 1 Report
Comments and Suggestions for Authors
All of my review comments have been addressed by the authors.
Author Response
We would like to thank the referees for their constructive and quality feedback helping to improve the article.
Reviewer 2 Report
Comments and Suggestions for Authors
The authors have well addressed all my concerns, no further comments.
Author Response

(The authors gave the same response as above.)

Reviewer 3 Report
Comments and Suggestions for Authors
Authors have addressed the comments in the revised paper. Paper is acceptable.
Author Response

(The authors gave the same response as above.)

Reviewer 4 Report
Comments and Suggestions for Authors
Figures should be placed as close as possible after their first mention and description in the text. Also, figures should be in the same section as the description, not before the description and in the previous section. Thus, Figures 1 and 2 should be moved from the introduction to Section 2, etc.
Author Response
We would like to thank the referees for their constructive and quality feedback helping to improve the article.
"Please see the attachment."
